# Effectiveness of Measures to Reduce the Influence of Global Climate Change on Tomato Cultivation in Solariums—Case Study: Crișurilor Plain, Bihor, Romania

Mihai Cărbunar [1], Olimpia Mintaș [2], Nicu Cornel Sabău [3,*], Ioana Borza [1], Alina Stanciu [1], Ana Pereș [3], Adelina Venig [3], Mircea Curilă [3], Mihaela Lavinia Cărbunar [1], Teodora Vidican [1] and Cristian Oneț [3]

1   Department of Agriculture-Horticulture, Faculty of Environmental Protection, University of Oradea, 26 Magheru Street, 410048 Oradea, Romania; mcarbunar@uoradea.ro (M.C.); iborza@uoradea.ro (I.B.); astanciu@uoradea.ro (A.S.); mihaelacarbunar@yahoo.com (M.L.C.); tvidican@uoradea.ro (T.V.)
2   Department of Animal Science and Agritourism, Faculty of Environmental Protection, University of Oradea, 26 Magheru Street, 410048 Oradea, Romania; olimpia.mintas@uoradea.ro
3   Department of Environmental Engineering, Faculty of Environmental Protection, University of Oradea, 26 Magheru Street, 410048 Oradea, Romania; peresana@uoradea.ro (A.P.); avenig@uoradea.ro (A.V.); mcurila@uoradea.ro (M.C.); cristian.onet@uoradea.ro (C.O.)
*   Correspondence: nsabau@uoradea.ro

**Abstract:** Tomatoes, one of the most appreciated vegetables consumed, are crops well adapted for cultivation in arid and semi-arid conditions, with the success of large yields guaranteed by covering water consumption through irrigation. Solar Pumps—SP are driven by Photovoltaic Panels—PV (SPAPV), eliminating the dependence on electricity or diesel; they are environmentally friendly because they generate carbon-free electricity, and the cost of operation and maintenance is lower. In order to preserve the water administered by drip to the tomato crop grown in solariums, mulching is used. In Husasău de Tinca, in the Crișurilor Plain, the cultivation of tomato varieties without mulching (WM) and with mulching with black foil (MBF) was studied. To answer the question "How effective are water conservation measures in terms of energy independence?", two variants of SPAPVs, direct pumping (ADP) and storage tank (AST) were simulated. It is proposed to determine the water consumption of tomatoes (ETRo), using the temperatures inside the solarium. In 2016, the average temperatures during the vegetation period with insurance of over 20%, were observed. The specific investment (SI) is 214,795 Euro ha$^{-1}$ in the case of ADP and respectively 202,990 Euro ha$^{-1}$ in the case of ATS. The payback period (IPT) is between 2.68 years and 2.53 years for the ADP variant and between 1.63 years and 1.54 years for the ATS variant. The indications for water use and irrigation water use show that in the MBF variant, the water administered by localized irrigation is better utilized than in the WM variant. The novelty of this study is the evaluation of the effectiveness of measures to reduce the effects of global climate change on tomatoes grown in solariums, useful for forecasting and watering restrictions, water consumption is determined from the air temperature measured inside the solarium, with insurance of over 20%. Taking into account the working hypotheses, the distribution of irrigation water in the solarium, with the help of SPAPVs, for tomatoes grown in the MBF variant, and a drip irrigation arrangement with ATS, the energy independence of the farm is ensured, the yields obtained are economically efficient, and the irrigation water is used rationally.

**Keywords:** tomatoes; drip irrigation; mulching; solar pump; photovoltaic panel; economic indices; irrigation water indices

## 1. Introduction

Global population growth and climate change are among the biggest challenges facing the world. Added to these are the health crisis, caused by the energy crisis, which is manifested by the continuous increase in energy prices [1–3]. The general effects of climate

change are visible through the negative influences on the water circuit in nature, manifested by changing the distribution, in time and space of the atmospheric water cycle, with clear trends to increase the coefficient of variation [4]. The increase of the coefficient of variation indicates both the increase of floods and droughts frequency, which are accompanied by an obvious tendency to increase the value of the temperature [5,6]. Due to their regularity, they lead to land degradation through desertification, with negative implications for the possibilities of providing water for industry, agriculture, and the supply of drinking water to the population [7,8].

The complex effects of climate change on agriculture are manifested by reducing water reserves in the soil, drastically reducing the accessibility of water to plants, declining agricultural production, and, thus, reducing the diversity, structure, composition, and distribution of crop plants [9–11]. It is estimated that the main climatic element responsible for the growth and production of crops is temperature, which together with rainfall influences evapotranspiration (ET). Through these two components (evaporation + perspiration), evapotranspiration is the way in which the water consumption of crops is measured [12–14]. The trend of evolution of water consumption of crops in arid conditions is increasing, a trend imposed by the evaporation component of consumption [15–17]. Under these conditions, the watering of crops is provided to a lesser extent by atmospheric precipitation, and it is, therefore, necessary to supplement it with irrigation [18–20]. Given that in times of drought, even the natural water reserve is limited, by principles of sustainable agriculture, it is necessary to use early warning systems for droughts and apply effective management [21–24]. Due to increasing water requirements for crops, efficient management of water resources in agriculture means efficient use of irrigation water, conservation of the soil water supply, and a reduction of energy consumption for irrigation [25–28].

The energy consumption in the agri-food sector is high, with high values of consumption, plant production, animal husbandry, and the transport of agricultural products and agricultural machinery, fertilizers, pesticides, etc. [29]. The structure of energy sources is very varied. In 2018, in EU countries, non-renewable energy produced by burning fossil fuels was still 60% [30]. If we analyze the plant production sector, we notice the preponderance of energy produced by burning liquid fossil fuels, derived from crude oil, for agricultural machinery involved in mechanized agricultural work, and high electricity consumption, necessary for pumping large volumes of water used for irrigation [31]. At the same time, the agricultural sector contributes 10–12% of its total greenhouse gas emissions annually [32]. Action must be taken to reduce the effects of global climate change and improve the quality of the environment by reducing greenhouse gas emissions in agriculture by requiring the gradual replacement of fossil fuel sources with energy from renewable sources, such as biomass, wind, or solar [33–35]. Given that in agriculture the highest consumption of electricity is used for pumping irrigation water, the use of solar energy to produce the necessary electricity has several advantages: high development potential; energy independence; low impact on the environment due to the reduction of greenhouse gas emissions; and economic efficiency [36].

Considering that the water pumping for irrigation is power consuming, since the 1970s, it has been proposed to use renewable sources of pumping energy, particularly photovoltaic energy [37]. Then came solar pumps (SP), pumps powered directly by electricity produced by photovoltaic panels (PV), with the use of water solar pump systems with photovoltaic panels (SPAPVs) seeing spectacular development [38]. In the absence of electricity supply networks, PV is a viable solution for agriculture, being cheaper than pumps powered by liquid fuel engines. The increase in oil prices and the reduction of PV production costs by 30–60% in the last 10 years, from 76 USD/Wat in 1977 to 0.30 USD/Wat in 2015, make the use of SPAPVs more attractive to decision-makers, technicians, and users [39].

SPAPVs used for water pumping have several remarkable advantages: they reduce or eliminate dependence on electricity or diesel; they are environmentally friendly because they generate carbon-free electricity; the cost of operation and maintenance is lower, which minimizes the tendency to reduce investment [40]. Water pumping with systems powered

by energy produced by PVs is used more and more for domestic water supply [41], for populations and animals in arid and semi-arid rural areas [42]. SPAPVs are effective solutions for irrigating isolated areas, with a convenient distribution of daily sunshine and global solar radiation [43]. SPs are driven by electric motors powered by PV. The electric motors used can run on direct current or alternating current. In the case of alternative current motors, it is necessary to add an inverter to the system construction, which converts the direct current produced by the solar panel into an alternating current. The efficiency of PVs depends on the duration of solar radiation and the ambient temperature, which can have negative influences. Besides, the performances of SPAPVs are directly influenced by the climatic conditions of the area [44]. Moreover, research on optimizing the pumped water flow aimed to establish the maximum power point of PVs, depending on the mode of coupling in series or in parallel, using modeling with the help of artificial neural networks [45]. In the case of mixed PV coupling, the best performances are obtained for solar pumps. Numerous researches on the architecture of SPAPV have shown that in order to improve the efficiency of water use, increase distribution uniformity and increase yield, at direct pumping systems (that pump water only when PV captures solar radiation) [46], water storage tanks [47] or accumulators (batteries) have to be added. The arrangements for localized irrigation with a water storage tank allow the administration of water to the plant when the energy generated by PV is not enough [48]. The SPAPV can be easily installed near the place of consumption, and the area occupied by PV can be optimized, which is why they are often used for irrigating tomatoes grown in solariums [49]. In addition, the use of SPAPVs for drip irrigation allows the complete automation of the warning and forecast of the application of the irrigation norms necessary to complete the soil moisture [50].

The main condition that an irrigation system fed by a SPAPV must meet is to ensure the optimal water consumption of the crop, which is the amount of water consumed by one hectare of the crop during the vegetation period, ensuring maximum yield [51]. Because the water requirement of a crop is composed of the amount of water lost from the soil reserve by evaporation and the amount of water consumed by the crop through perspiration, for consumption, the notion of evapotranspiration (ET) is used [52]. A lot of methods for determining the water consumption of crops (evapotranspiration of crops) are presented in the specialty literature, each of them being adapted to the purpose (determining the need for irrigation water), natural conditions in the analyzed area, endowment with specialized equipment, and last but not least, the available climate database [53,54]. Depending on how the ET is evaluated, the literature mentions potential evapotranspiration (PET), the reference evapotranspiration (ETRo), and the actual evapotranspiration (ETa) [55]. PET is the rate of water transfer from moist soil by evaporation to the surface and the transpiration of plants, in conditions of balance between the water reserve in the soil, the water consumption of the crop, and the atmospheric conditions. PET has a more general, ambiguous interpretation, being the amount of water that can be consumed from the soil reserve under ideal conditions. [56]. ETRo refers to the amount of water lost through evaporation and perspiration of a grassy surface, maintained at a low height, considered the reference crop. ETRo eliminates PET-related ambiguities and allows a more realistic characterization of the effect of microclimate on the evaporative transfer from the soil-plant system to the atmosphere, above a cultivated area [57]. ETa represents the current water consumption of crops, determined for a certain period of time, in experimental fields, by calculating the water balance in the soil [58].

For plants grown in the open field, part of the water consumption is covered by rainfall and the water reserve in the soil, however, in the case of indoor crops, the rainfall is missing, which is why ET must be covered by irrigation. In the case of SPAPV used for irrigation in field crops, they have the advantage that in rainy periods, when the sky is overcast, their operation is not necessary. Water pumping is necessary in periods without precipitation, with clear skies, when global solar radiation is maximum. On the contrary, in the case of the irrigation of plants grown indoors, SPAPV must also pump water during periods of cloudy skies [40].

Tomatoes (*Solanum lycopersicum* L. *Lycopersicon esculentum* Mill), due to their low caloric value and high vitamin content, are among the most popular fresh vegetables, which is why all over the world, there is interest in increasing production. Moreover, organic tomatoes grown in solariums, in the current global climate change, present a great challenge in terms of water consumption management and economic efficiency. Under these conditions, especially in arid and semi-arid areas, water becomes a limited economic resource, the largest consumer being agriculture, which uses it mainly for crop irrigation. Efficient use of water for irrigation, in order to achieve the highest possible yields and to save water, requires the use of efficient irrigation methods, such as localized irrigation, avoidance of water loss, warning, and scheduling of irrigation management [59].

The water consumption of tomatoes is very different, depending on the type of crop (field or solarium) and the climatic conditions of the year considered. These differences may also be due to the fact that the design of irrigation systems uses precipitation, with a 20% non-exceedance insurance, recorded over a period of about 30 years. The assurance of not exceeding the value Xi, from a series of observations Xn, indicates the frequency of the years in which the observed values will be lower than the considered value Xi. If we take into account the fact that under controlled conditions, precipitation does not participate in supplementing crop consumption and that the recorded temperatures are those used to determine the ET, it is proposed to use the average annual temperature with 20% insurance [56].

Due to the current global climate change, the rising price of electricity, and the fact that the irrigation of tomatoes in Husasău de Tincă is done by drip using a pump powered by electricity from the national grid, the aim is to achieve energy independence by using the SPAPVs. Given that water consumption of tomatoes grown in solariums is expected to increase and that alternatives to water conservation from the soil by mulching have been tested, it is interesting to assess the effectiveness of the means f reducing the effects of climate change on tomatoes grown in solariums?

The main objective of this paper is to use the guided simulation of an operational SPAPV (with direct pumping—ADP and with a water storage tank—ATS) from Husasău de Tinca, to highlight the efficiency of some soil-water conservation methods (mulching with black foil—MBF) in the conditions of the Crișurilor Plain. In order to achieve this objective, a number of economic indicators (The surface that can be cultivated, Specific investment—SI, Investment payback time—IPT), and indicators of water use (Water Use Coefficient—WUC; Irrigation Water Use Coefficient—IWUC; Water Use Efficiency—WUE; Irrigation Water Use Efficiency—IWUE) have been determined.

In order to achieve the objectives set out above, some hypotheses have been considered. Irrigation water taken from a surface source (watercourse) covers the water consumption of tomatoes [60,61]. Because watering is applied before the tomatoes are planted, the water supply in the soil, at the beginning of the vegetation period (May 1), is considered to be halfway through the active humidity range, between Field Capacity (FC) and Minimum Ceiling (MC) [62,63]. During periods of maximum tomato consumption, if several waterings are scheduled in one day, some waterings may be administered in advance or one day late [64].

## 2. Materials and Methods

### 2.1. Location of the Experimental Field from Husasău de Tinca, Bihor County

The present case study is located in the Crișurilor Plain in Husasău de Tinca, Bihor County, where there are 4 solariums arranged for drip irrigation, in which the influence of foil type and mulching on tomato production was studied (Figure 1).

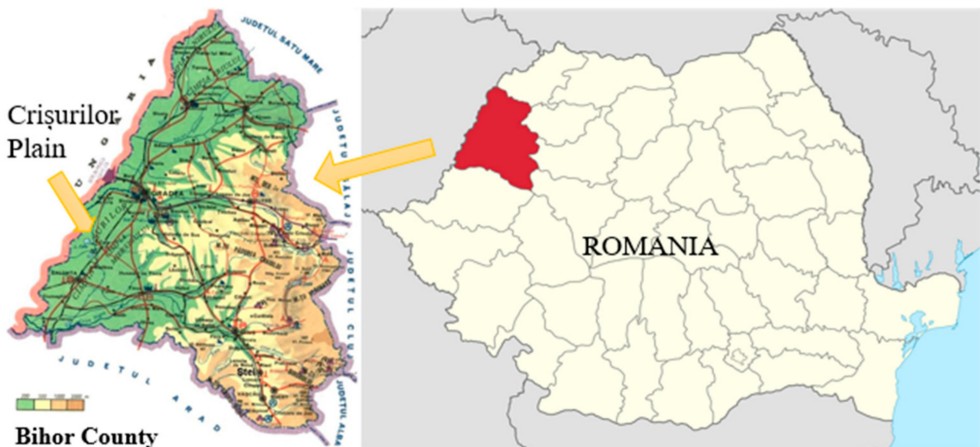

**Figure 1.** Location of the experimental field from Husasău de Tinca, Bihor County, Romania [65,66].

To classify the climatic conditions from Husasău de Tinca, recorded data for 48 years (1970–2018) was used (precipitation, air temperature, relative humidity, speed of wind, and duration of sunlight) from the Oradea Meteorological Station located at approx. 30 Km (Table 1) [67].

**Table 1.** Geographical coordinates of the Husasău de Tinca Experimental Field and of the Oradea Meteorological Station.

|  | Latitude | Longitude | Altitude (m) |
|---|---|---|---|
| Husasău de Tinca | 46°49′10″ N | 21°54′39″ E | 134 |
| Oradea | 47°02′20″ N | 21°53′58″ E | 132 |

From an agricultural point of view, this period is defined by average rainfall values of 620.0 mm, with a variation range between 411.0 mm and 889.8 mm, and average air temperature values of 10.7 °C, with a minimum of 8.9 °C and a maximum of 12.45 °C (Table 2). In order to assess the suitability of using PV for SPAPV supply, it should be noted that the average number of sunny days is 296.56 and the average annual sunshine duration is 2104.43 h.

**Table 2.** Climatic data—average values of the agricultural years (October–September) at the Oradea Meteorological Station (1970–2018).

| Climatic Data | Number of Agricultural Years | MEAN | MIN | MAX | STDEV * | MSE ** |
|---|---|---|---|---|---|---|
| Rainfall (mm) | 48 | 620.0 | 411.0 | 889.8 | 120.3 | 17.4 |
| Air temperature (°C) | 48 | 10.70 | 8.90 | 12.45 | 1.23 | 0.03 |
| Relative humidity (%) | 48 | 76.68 | 67.50 | 83.58 | 3.61 | 0.52 |
| Speed of wind (ms$^{-1}$) | 48 | 2.93 | 2.40 | 3.60 | 0.30 | 0.04 |
| Number of sunny days | 48 | 296.56 | 271.00 | 321.00 | 10.88 | 1.57 |
| Duration of sunlight (h) | 48 | 2104.43 | 1814.4 | 2535.2 | 139.24 | 20.1 |

* STDEV—Standard deviation; ** MSE—Mean square error.

In solariums, the soil type is a Haplic Luvosoil one, with average colloidal clay content (<0.002 mm) on the watering depth of 50 cm is 34.2%, a bulk density (BD) of 1.48 g cm$^{-3}$, a field capacity (FC) of 24.0%, and a wilting coefficient (WC) of 9.7%.

The two varieties of tomatoes grown in solariums in Husasău de Tinca are one grown without mulch (WM) and one mulched with black foil (MBF). In the first case, the distribution of water to the plant is conducted with T-type drippers, while for the second one, it is undertaken with microtubes (Figure 2).

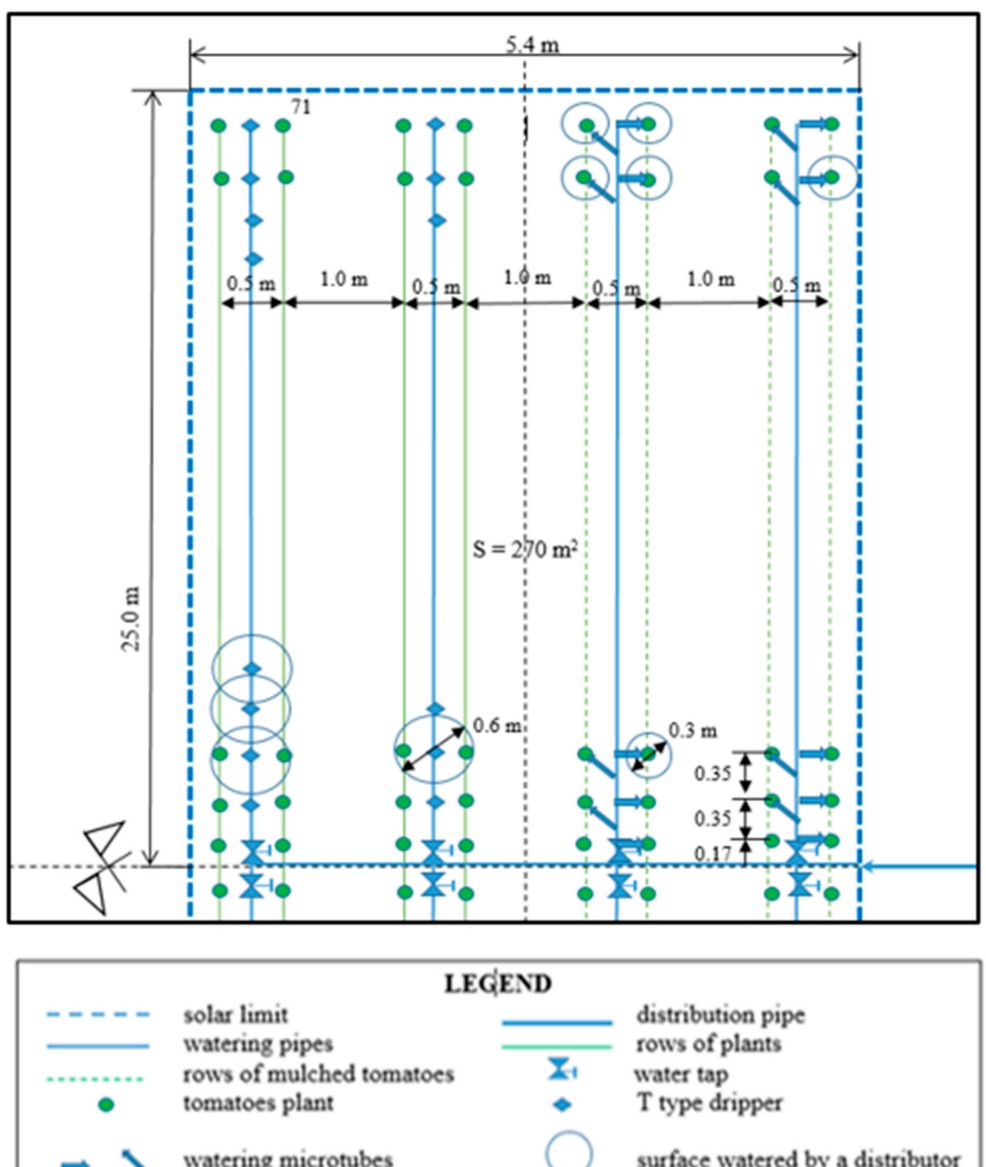

**Figure 2.** Water distribution of tomato variant located in Husasău de Tinca solariums.

The experimental field was arranged in subdivided blocks, with two variants (WM and MBF) in 4 random repetitions. Considering that the placement of the repetition plots was in Latin square, the yields obtained in the period 2015–2017 were statistically processed using the appropriate methodology and the Excel program [68]. The yields differences of the studied variants were made with the Fischer test (F) and the Student test (*t*), in order to establish the statistical significance.

### 2.2. Water Consumption of Tomatoes Grown in Solariums in Husasău de Tinca

For the design of the SPAPV to provide drip irrigation in the four solariums, it was necessary to determine the need for water, starting with the determination of the ET of tomatoes grown indoors.

To determine the water consumption of tomatoes grown indoors, we used indoor temperatures recorded in the growing season from a year in which there was a 20% assurance of exceeding the air temperature. The agricultural year for which the average outside temperature of the vegetation period (April–September) exceeded the insurance of 20% was established using the data recorded at the Oradea Meteorological Station in the

period 1970–2018. The Pearson type III function [69] was used to determine the probability of exceeding the air temperature using Equation (1):

$$\varphi(u) = A \cdot u \cdot e^{a-u}; \tag{1}$$

where:

$$u = -\frac{x}{2 \cdot \sigma^2}; \text{and } A = \frac{1}{2 \cdot \sigma^2 \cdot \Gamma_{(a+1)}}; \tag{2}$$

In which:

X—the value of a series of observations for which the probability is to be determined;

$e$—the basis of natural logarithms;

$a$—function parameter;

$\sigma$—the mean square deviation of all the values of the studied event;

$\Gamma_{(a+1)}$—Euler integral (second case) or Gama function; The probability of exceeding a value $x_i$ from a string $x_1, x_2 \ldots x_n$ is obtained by integrating the curve given by the considered function, on the interval $x_1$–+∞ (3):

$$P = \int_{x_1}^{\infty} \varphi(x) dx; \tag{3}$$

In which:

P—Probability of exceeding the value $x_i$;

$\varphi(x)$—the function that describes the evolution of event x for the observation period;

After the sequence of average values of the air temperature in the warm season of each agricultural year: $x_1, x_2 \ldots x_n$, has been ordered in descending order, we have calculated for each value the probability P(%) using Equation (4):

$$P(\%) = \frac{i}{n+1} \cdot 100; \tag{4}$$

where:

i—the sequence number of the $x_i$ value, in the descending sequence;

n—the total number of values ($x_n$) of the analyzed string;

Chosen from this table was the agricultural year and respectively the average temperature of the warm season corresponding to the insurance of exceeding 20%.

Considering the average monthly values from the vegetation period, with the assurance of 20% as external values (Ta) of the solariums from Husasău de Tinca, these were transformed into internal temperatures ($T_i$), using the correlative link (5) established by previous researches [70].

$$T_i = 0.8796 \cdot Ta + 6.281; \tag{5}$$

where:

$T_i$—Average monthly temperature inside solariums (°C);

Ta—The average monthly temperature outside the solariums, determined in Oradea, with the insurance of 20% (°C);

With their help, the potential evapotranspiration (PET) was determined using the Thornthwaite Equation (6) and the DrinC program [71,72].

$$PET = 1.6 \cdot \left(\frac{10 \cdot t}{I}\right)^a \cdot K; \tag{6}$$

where:

PET—the potential evapotranspiration (cm);

t—average inside temperature of the month from the vegetation period (°C);

I—annual heat index, calculated as the sum of the 12 monthly indices i (7):

$$I = \sum_1^{12} i; \quad i = \left( \frac{t_n}{5} \right)^{1.541};$$

(7)

monthly indices are calculated only for positive temperature.

$t_n$—normal monthly average temperature (°C);

a—an empirical coefficient, determined by Equation (8):

$$a = 0.000000675 \cdot I^3 - 0.000771 \cdot I^2 + 0.01792 \cdot I + 0.49239;$$

(8)

K—monthly PET correction coefficient according to the length of the day, determined according to the latitude of the considered area.

In order to obtain the monthly water requirement of tomatoes grown in solariums (ETRo), the calculated PET data must be converted using the correction coefficients specific to the crop considered [73]. The culture coefficients Kc, for the tomatoes grown in solariums at Husasău de Tinca, were determined experimentally, both for the WM variant and for the MBF variant [74].

*2.3. Irrigation Regime of Tomatoes Grown in Solariums in Husasău de Tinca*

The watering norm (*m*) represents the amount of water administered by irrigation for one hectare of a certain crop. The amount of water administered during one watering depends on the hydrophysical characteristics of the soil and the water storage capacity in the soil. The relationship used to determine the watering rate (*m*) with drip irrigation (9), takes into account that the moistened surface is reduced compared to sprinkler watering, due to the fact that only the root area of the plants is moistened [51,74].

$$m = \frac{100 \cdot H \cdot BD(FC - WC)}{\eta} \cdot y \cdot \frac{P}{100};$$

(9)

where:

*m*—norm of water application by drip irrigation (m$^3$ ha$^{-1}$);

*H*—depth of soil wetting (m);

*BD*—volumetric weight or bulk density of the soil (t m$^{-3}$ or g cm$^{-3}$);

*FC*—field capacity for water of the soil (%);

*WC*—wilting coefficient (%);

*y*—the fraction in the range of available moisture content (*FC-WC*) to be filled with water (easily accessible water between FC and MMC—minimum moisture content for watering);

*P*—the percentage of soil surface actually moistened;

*η*—the efficiency of the uniformity of the watering along the watering pipe (0.8–0.9).

Considering that the density is 5.7 plants m$^{-2}$, and the distribution of water per plant for the WM variant is done with T-type tubes, the percentage of the moistened surface was 43.6% for the MBF variant, and with microtubes, the percentage was 40.5%.

The irrigation regime during the vegetation period of the tomatoes establishment starts from the watering norm m, and includes the number of waterings necessary each month to cover their optimal consumption [74,75]. To determine when to apply a new watering we started from the assumption that the initial water reserve in the soil at the beginning of the vegetation period (1 May) is halfway through the active humidity range, which is the difference between the field capacity (FC) and minimum ceiling (MC) [76]. Knowing the average monthly daily consumption of tomatoes, we can determine when the soil moisture reaches the MC and when is necessary to supplement the soil moisture by applying a new watering norm [77]. This hypothesis, in field conditions, is being satisfied in rainy springs, but in solar conditions, to ensure soil moisture at planting, a supply of watering is applied outside the vegetation period [63].



### 2.4. Design of SPAPV from Tomatoes Grown in Solariums at Husasău de Tinca

The main characteristics required for the design of water SP are the pumped flow and the required pressure in the pipes [78]. Hydraulic pressure losses or pressure losses are the energy lost by the water when passing through the pipes due to friction within them. Their knowledge is important to determine the pumping height at discharge, which is necessary for the proper functioning of the drippers [79]. When estimating the pressure losses, variants of SPAPV arrangement were simulated, using direct pumping (ADP) of water (Figure 3a) and with gravitational distribution using a water storage tank (ATS) (Figure 3b).

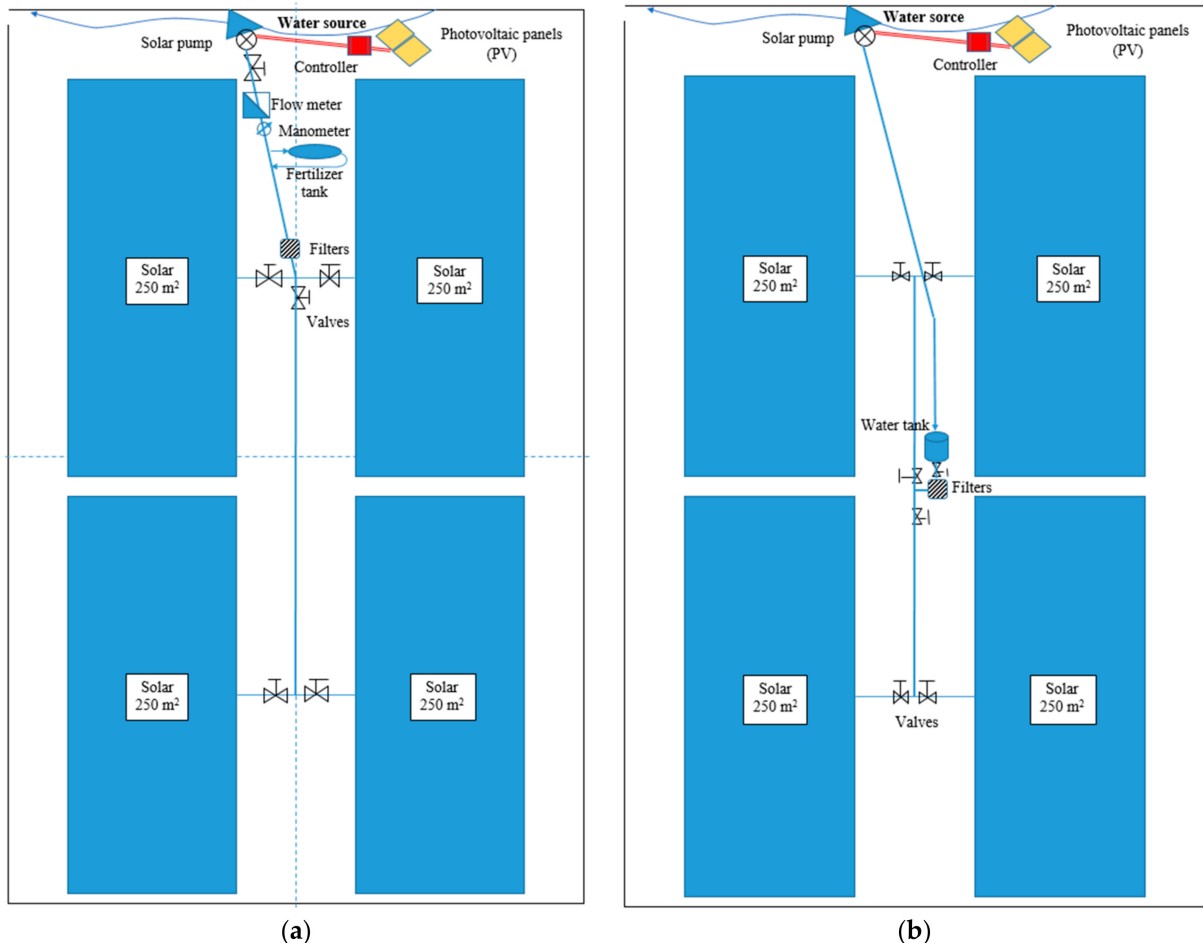

**Figure 3.** Arrangements for localized irrigation of solariums: (**a**) with direct pumping (ADP); (**b**) with a water storage tank (ATS).

The solar pump (SP) used must provide a pressure of 0.2 bar, i.e., 2 m of water column (MWC) required for the operation of the drippers. Given that the water source is a surface one, to this is added the suction height of 1.0 m and the hydraulic load losses on the transport and local pipes, estimated at approx. 1.0 MWC. Therefore, for our conditions, an SP with a DC motor is required to ensure a pumping height of at least 4 MWC. An SP that provides this pressure can also pump water into the compensation tank located on a support, at a minimum height of 2.5 m. Considering that most SPs are depth pumps designed to pump water from boreholes, and have high pumping heights but low pumping flows, several offers of the most well-known manufacturer of SPs, the German company Lorentz, were analyzed (Table 3).

**Table 3.** Variants of solar pumps produced by Lorentz.

| Manufacturer | Type | Pumping Height (MWC) | Debit (m$^3$ h$^{-1}$) | Rated Power (kW) | Reference |
|---|---|---|---|---|---|
| Lorentz | PS 2-600CS-17-1 | 12 | 18 | 0.7 | [80] |
| Lorentz | PS2-1800CS-37-1 | 14 | 36 | 1.8 | [80] |
| Lorentz | PS-150 BOOST-330 | 50 | 1.3 | 0.3 | [80] |
| Lorentz | PS600 BADU Top12 | 4 | 10 | 0.5 | [81] |

The PS600 BADU Top12 pump was chosen from the analyzed SP variants, because it had the lowest pumping height and came with a PS 600 controller. The SP is driven by an ECDRIVE 600 BADU Top DC motor, driven by the current produced by solar panels (340–900 Wp) via a controller (Figure 4). The water flow that was pumped during different periods of the growing season was estimated using the characteristic flow curve of the chosen pump with power supply (W) and the required pumping height (Figure 5).

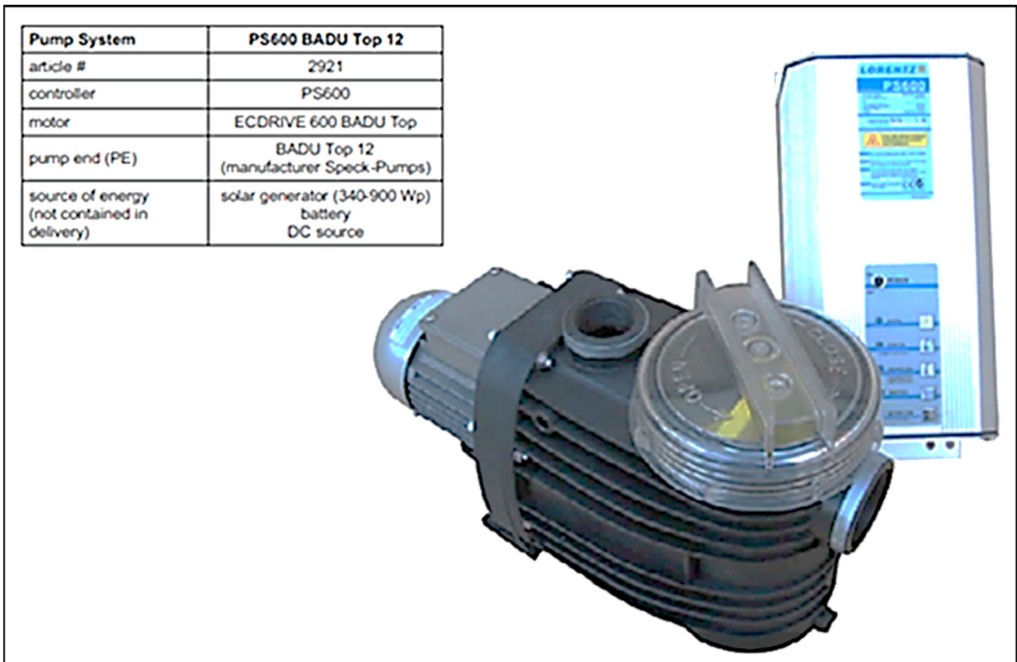

**Figure 4.** Technical characteristics of the SP PS600 BADU top 12 [81].

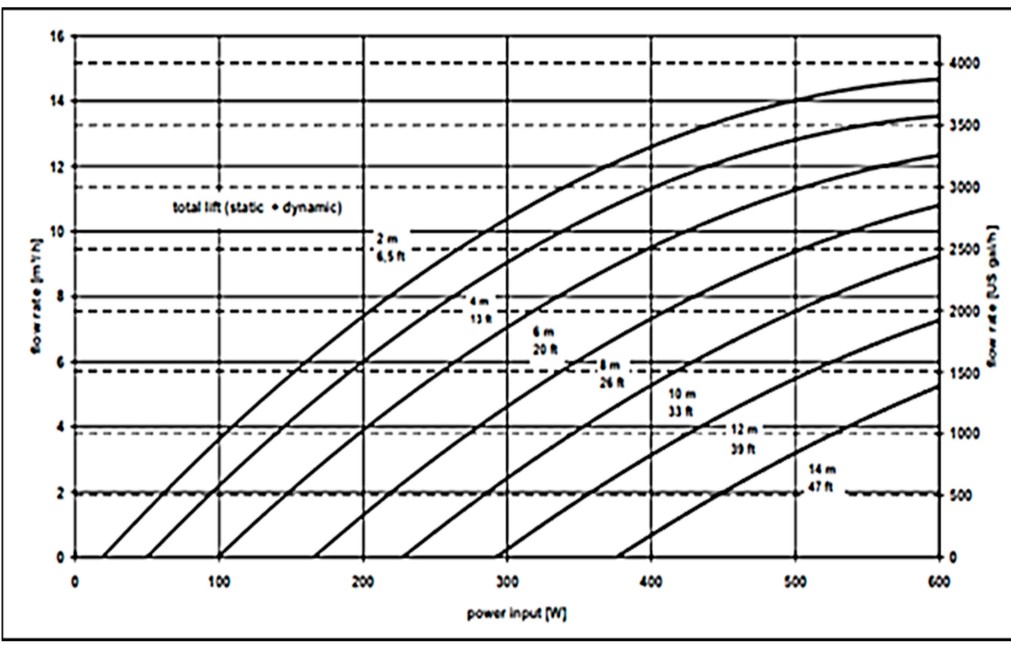

**Figure 5.** Flow characteristic curve—input power and required pumping height [81].

The Angstrom-Prescott Equation (10), was used to estimate the average daily global solar radiation H ($Whm^{-2}$), with the following form [82]:

$$H = Ho \ (0.2881 + 0.7429 \ \sigma \ 0.6168) \tag{10}$$

where:

Ho is solar irradiation in clear sky conditions;

$\sigma$ is the relative daily duration of sunshine;

The monthly values of horizontal and inclined global solar irradiation at 45° for Husasău de Tinca, Crișurilor Plain, were estimated using the PVGIS-5 temporal irradiation database [83]. To determine the average daily and hourly values used, data registered at the Oradea Meteorological Station, Bihor County was used including the number of days (monthly and annually) with the sun, the duration of sunshine in hours (monthly and annual amounts), and the duration of the solar day in hours [84].

Photovoltaic panels (PV) were chosen from a wide range of types and features on the market. Five types of PV produced in countries with a tradition in this field were analyzed (Table 4).

**Table 4.** Variants of photovoltaic panels (PV).

| Trade Name | Manufacturer | Type * | Surface ($m^2$) | Power (W) | Efficiency (%) | Source |
|---|---|---|---|---|---|---|
| SPM041751200 | Victorian Energy, The Netherlands | M | 0.99 | 175 | 13.0 | [85] |
| TM-P660265 | Tamesol, Girona, Spain | P | 1.46 | 265 | 14.4 | [86] |
| SGSP-150 | Sun Power, Shenzhen, China | M | 0.78 | 150 | 22.6 | [87] |
| FVG-185M-MC | FVG Energy, Cittadella, Italy | M | 1.28 | 185 | 14.5 | [88] |
| AE320M6-60 | AE Solar, Königsbrum, Germany | M | 1.66 | 320 | 19.3 | [89] |

* M—Monocrystalline; P—Polycrystalline.

The AE320M6-60 PV, produced in Germany, was preferred due to the high power supplied of 320 W at a good efficiency of 19.3%.

*2.5. Effectiveness of Measures to Reduce the Effects of Climate Change and Energy Consumption*

For the evaluation of the economic efficiency of the SPAPVs, for the two possibilities of arrangement for drip irrigation: ADP and ATS; the specific investment (SI) was determined. SI represents the cost of arranging one hectare of land for drip irrigation, including the cost of materials (SP, controller, PVs, pipes, valves, manometers, fertilizer tank, filters, water storage tank), labor (cost of construction work), and possibly the rates owed to the bank for the loan.

The investment payback time (IPT) represents the number of years in which the expenses associated with the two types of arrangement for drip irrigation (ADP and ATS) are recovered by the annual net profit (ANP) obtained, as a result of capitalizing on the obtained yield. To determine the IPT expressed in years, Equation (11) was used [90,91]:

$$IPT(years) = \frac{SI}{ANP}; \tag{11}$$

ANP represents the difference between the income realized by capitalizing the annual yield (IY) and the production expenses (PE), represented by materials (seeds, black foil, treatments, etc.), salaries, taxes, and duties.

$$ANP = IY - PE; \tag{12}$$

The annual yield of certified organic tomatoes, sorted by 2 quality categories was capitalized on for the agri-food market in Oradea.

For calculating the irrigation water indices, Water Use Coefficient (WUC), Water Use Efficiency (WUE), Irrigation Water Use Coefficient (IWUC), and Irrigation Water Use Efficiency (IWUE) were used.

WUC is the ratio of water consumption (ETRo) to crop yield (Y) expressed in $m^3 \ kg^{-1}$ [61]:

$$WUC = \frac{ETRo}{Y}; \tag{13}$$

WUE ($kg \ m^{-3}$) is the inverse ratio of WUC, given by the crop yield for the unit of water consumed [70]:

$$WUE = \frac{Y}{ETRo}; \tag{14}$$

IWUC and IWUE are the ratios between the irrigation norm ($\Sigma m$) during the growing season and the increased yield (IY), brought about by the application of irrigation [92,93].

$$IWUC = \frac{\Sigma m}{IY} \ and \ IWUE = \frac{IY}{\Sigma m}; \tag{15}$$

Given that in our case, we do not have tomato productions for the without irrigation version, and the production increases cannot be calculated, for the calculation of IWUC and IWUE we used yields (Y) from the irrigated variants.

## 3. Results

*3.1. Irrigation Regime of Tomatoes Grown in Solariums*

The agricultural year with the average air temperature (12.2 °C) having the insurance of exceeding 20%, at the Oradea Meteorological Station was 2015–2016. For the vegetation period of tomatoes grown in solariums, the average air temperature was 18.8 °C, while inside solariums it was 22.8 °C. PET registers monthly values ere between 75.31 mm in April and 171.46 mm in July (Table 5).

**Table 5.** Water consumption of tomatoes (ETRo) grown in solariums in Husasău de Tinca, during the vegetation period of the agricultural year 2015–2016.

| Specification | U.M. | Vegetation Period | | | | | | Average Sum |
|---|---|---|---|---|---|---|---|---|
| | | April | May | June | July | August | September | |
| Air temperature Oradea (Ta) | °C | 13.4 | 16.4 | 21.3 | 22.5 | 21.1 | 18.0 | 18.8 |
| Air temperature inside solarium (Ti) | °C | 18.1 | 20.7 | 25.0 | 26.1 | 24.8 | 22.1 | 22.8 |
| Potential Evapotranspiration inside solarium (PET) | mm month$^{-1}$ <br> m$^3$ ha$^{-1}$ | 75.31 <br> 753.1 | 110.37 <br> 1103.7 | 157.73 <br> 1577.3 | 171.46 <br> 1714.6 | 143.80 <br> 1438.0 | 99.34 <br> 993.4 | 7580.01 <br> 75,800.1 |
| The culture coefficient (Kc) WM variant | | 1.01 | 1.03 | 1.02 | 1.03 | 0.77 | 0.57 | 0.819 |
| The culture coefficient (Kc) MBF variant | | 1.00 | 1.02 | 1.00 | 0.96 | 0.72 | 0.55 | 0.875 |
| Optimal actual evapotranspiration (ETRo) WM variant | m$^3$ ha$^{-1}$ <br> m$^3$ ha$^{-1}$ day$^{-1}$ <br> mm day$^{-1}$ | 760.6 <br> 25.35 <br> 25.4 | 1136.8 <br> 36.67 <br> 36.7 | 1608.8 <br> 53.63 <br> 53.6 | 1766.0 <br> 46.97 <br> 47.0 | 1107.3 <br> 35.72 <br> 35.7 | 566.2 <br> 18.87 <br> 18.9 | 6945.7 <br> 217.21 <br> 217.3 |
| Optimal actual evapotranspiration (ETRo) MBF variant | m$^3$ ha$^{-1}$ <br> m$^3$ ha$^{-1}$ day$^{-1}$ <br> mm day$^{-1}$ | 753.1 <br> 25.10 <br> 25.1 | 1125.8 <br> 36.32 <br> 36.3 | 1577.3 <br> 52.28 <br> 52.6 | 1646.0 <br> 53.10 <br> 53.1 | 1035.4 <br> 33.40 <br> 33.4 | 546.4 <br> 18.21 <br> 18.2 | 6594.0 <br> 215.41 <br> 218.7 |

ETRo for the WM variant indicates a water consumption of 6945.7 m$^3$ ha$^{-1}$, higher than the MBF variant with consumption of 6594.0 m$^3$ ha$^{-1}$. The average daily consumptions of the two variants was very close, having values between 25.4–53.6 mm day$^{-1}$ for WM and respectively 25.1–53.1 mm day$^{-1}$ for MBF. The amount of water that could be administered in a watering (watering norm m), at a depth of 50 cm soil, and the surface actually watered without exceeding the water storage capacity in the soil was 182 m$^3$/ha for the WM variant and 169 m$^3$/ha for MBF (Table 6).

**Table 6.** Calculation of the watering norm (m).

| Specification | Depth H (m) | Bulk Density BD (g cm$^{-3}$) | FC-MC (%) | Percentage of Watered Area P/100 | Watering Efficiency η | Watering Norm (m) | | |
|---|---|---|---|---|---|---|---|---|
| | | | | | | m$^3$ ha$^{-1}$ | L m$^{-2}$ | L Plant$^{-1}$ |
| WM variant | 0.50 | 1.48 | 4.8 | 0.436 | 0.85 | 182 | 18.2 | 3.2 |
| MBF variant | 0.50 | 1.48 | 4.8 | 0.405 | 0.85 | 182 | 16.9 | 3.0 |

For the same water storage conditions in the soil, the lower values of m corresponding to the MBF variant, were due to the lower percentage of wetland than to the WM variant. Taking into account the fact that a distributor (dripper) was placed on each plant, it administered 3.2 L plant$^{-1}$ in the WM variant and 3.0 L plant$^{-1}$ in the MBF variant, respectively. For this, the T-type tubes at a pressure of 0.4 bar with an average flow of 1.25 L h$^{-1}$ must operate for 4 h, and the microtubes operated with a flow of 0.5 L h$^{-1}$ for 6 h.

The irrigation regime of tomatoes grown in solariums differed depending on the cultivation system practiced. The highest monthly watering norms were met in June, at 1638 m$^3$ ha$^{-1}$ for the WM variant, and in July, at 1690 m$^3$ ha$^{-1}$ for the MBF variant (Figure 6).

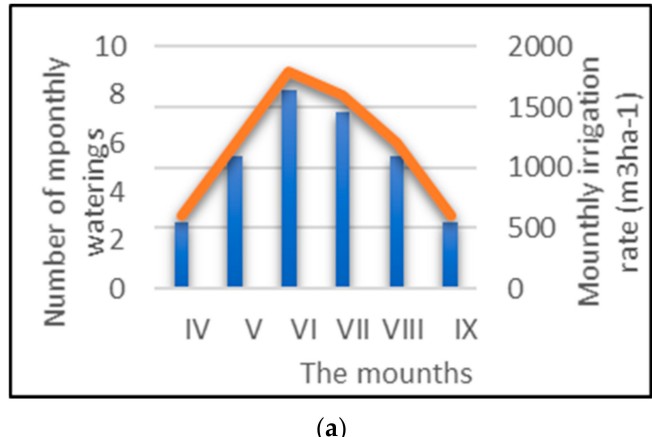
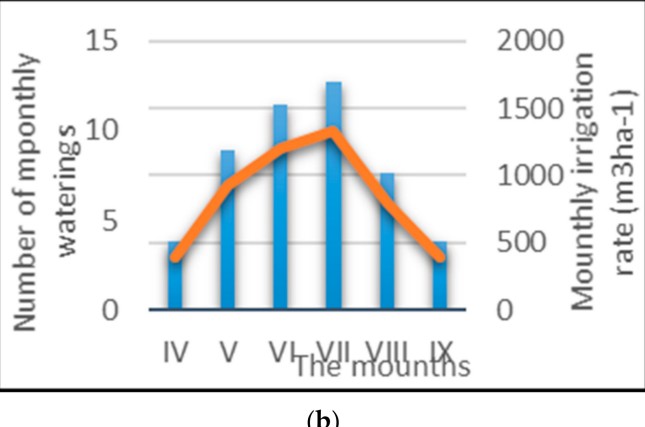

| (a) | (b) |

**Figure 6.** Irrigation regime of tomatoes grown in solariums: (**a**) without mulching (WM); (**b**) mulching with black foil (MBF).

If in the case of WM, 35 waterings totaling 6370 m$^3$ ha$^{-1}$ of water were required, for MBF, although the consumption was slightly lower, due to the fact that the water storage capacity in the soil was lower, 38 waterings using 6422 m$^3$ ha$^{-1}$ were required. Analyzing the data for the application of watering during the vegetation period of tomatoes grown in solariums in 2016 (the year with the insurance of exceeding temperatures of 20%) it was noted that, in most cases, the data were different for the two variants (Table 7).

**Table 7.** Watering scheduling dates for tomatoes grown in solariums.

| Months | Without Mulching (WM) | | | | Mulched with Black Foil (MBF) | | | |
|---|---|---|---|---|---|---|---|---|
| | **Data** | **n** | **m** | **Σm** | **Data** | **n** | **m** | **Σm** |
| April | 14, 22, 29 | 3 | 182 | 546 | 14, 20, 27 | 3 | 169 | 507 |
| May | 4, 9, 14, 19, 24, 29 | 6 | 182 | 1092 | 3, 7, 12, 16, 21, 26, 30 | 7 | 169 | 1183 |
| June | 2, 5, 9, 12, 15, 18, 22, 26, 30 | 9 | 182 | 1638 | 3, 6, 9, 12, 16, 19, 22, 25, 28 | 9 | 169 | 1521 |
| July | 3, 7, 10, 14, 18, 22, 26, 30 | 8 | 182 | 1456 | 2, 5, 8, 11, 14, 17, 21, 24, 27, 30 | 10 | 169 | 1690 |
| August | 4, 9, 14, 19, 24, 29 | 6 | 182 | 1092 | 4, 9, 14, 19, 24, 29 | 6 | 169 | 1014 |
| September | 7, 16, 26 | 3 | 182 | 546 | 7, 16, 25 | 3 | 169 | 507 |
| TOTAL | | 35 | | 6370 | | 38 | | 6422 |

Note: n—number of watering norms; m—watering norm (m$^3$ ha$^{-1}$); Σm—irrigation norm (m$^3$ ha$^{-1}$).

From the point of view of the number of waterings required in a month, it was observed that in July, 8 waterings times were required for the WM version, while 10 waterings times were required for the MBF version. The average return time with watering on the same surface was in the first case, 3.9 days and in the second case, 3.1 days.

### 3.2. The Main Characteristics of SPAPV from Tomatoes Grown in Solariums

Horizontal global solar irradiation in the calendar year 2016 for Husasău Tinca was 1319.1 kWh m$^{-2}$ with a maximum value estimated in July (189.8 kWh m$^{-2}$) and a tilt to 45° of 1641.4 kWh m$^{-2}$, the monthly maximum being 196.16 kWh m$^{-2}$ estimated in August (Figure 7). For the vegetation period of tomatoes grown in solariums (April–September), the horizontal solar irradiation accumulated was 1005.3 kWh m$^{-2}$, while in the case of PV inclined at 45°, it was estimated at 1038.9 kWh m$^{-2}$ (Table 8).

The average hourly value using the solar for days in the growing season iswas 391.8 W m$^{-2}$, with monthly values between 342.4 W m$^{-2}$ in June and 438.2 W m$^{-2}$ in August.

The power supply of the SP preferred, based on its technical characteristics, was the PV produced in Germany, AE320M6-60. Ideally, two panels could provide 640 W of power, sufficient for SP operation. By assessing the efficiency of transformation of solar irradiation

into electricity and the characteristics of the SP which indicated the need for peak power (Wp) over 340 W, the requirement for the PV was determined (Figure 8).

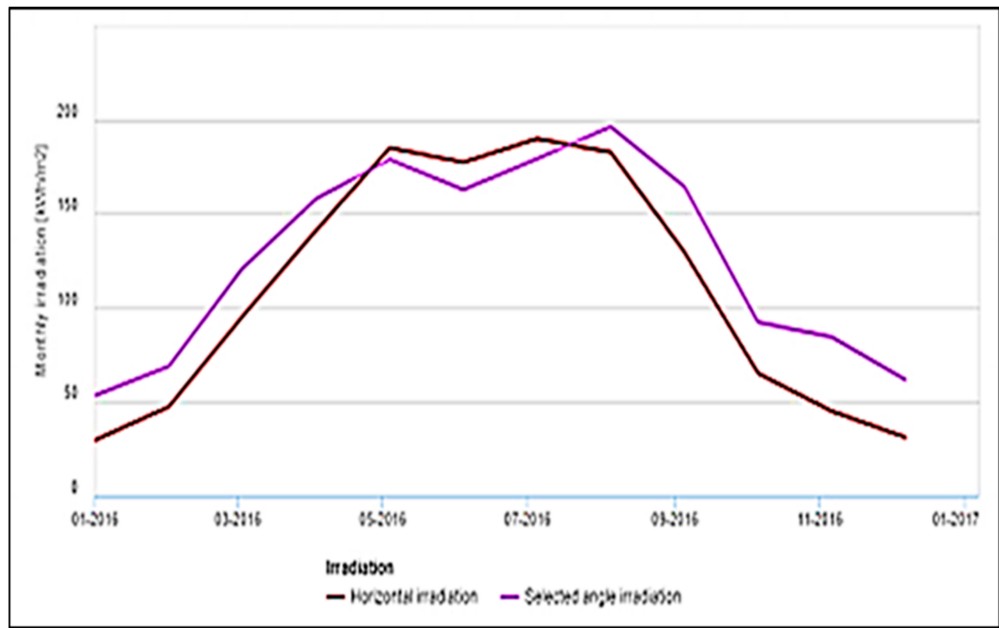

**Figure 7.** Estimation of monthly solar irradiation (global horizontal irradiation) for Husasău de Tinca, Romania, 2016 [83].

**Table 8.** Average global solar irradiation at 45°, Oradea Meteorological Station, 2016.

| Specification | April | May | June | July | August | September | Warm Season | Annual |
|---|---|---|---|---|---|---|---|---|
| Monthly solar irradiation—H (kWh m$^{-2}$) | 157.9 | 178.7 | 162.7 | 162.9 | 196.2 | 164.3 | 1038.9 | 1641.4 |
| Daily solar irradiation—(kWh m$^{-2}$) | 5.263 | 5.765 | 5.423 | 5.777 | 6.213 | 5.477 | 5.677 | 4.485 |
| Solar day (h) | 13.70 | 8.54 | 8.17 | 9.82 | 9.87 | 7.70 | 8.62 | 7.07 |
| Average duration of sunlight (h) | 7.36 | 8.54 | 8.17 | 9.82 | 9.87 | 7.70 | 8.62 | 7.07 |
| Hourly global solar irradiation—H (W m$^{-2}$) | 384.2 | 380.8 | 342.4 | 372.7 | 438.2 | 435.4 | 391.8 | 365.2 |

Due to the fact that for the power supply of SP the number of solar panels must be a multiple of two, it was concluded that 4 PVs were required, mounted inclined at 45°, which would produce over 438.8 Wh. The annual average of the produced electricity iswas 468 W, and that of the vegetation season was 502 W.

The PS600 BADU Top12 solar pump, produced by Lorentz, was chosen to equip the SPSAV, which providesd an average flow rate of 10 m$^3$ h$^{-1}$ at a maximum pumping height of 4 m. According to the characteristic curve of the SP, it was noted that in conditions of a lower pumping height and a supplied power greater than about 350 W, the supplied flow can increase up to 13.7–13.8 m$^3$ h$^{-1}$ (Figure 5). The water flow pumped by SPAPV is an average daily value, due to the fact that it was appreciated with the help of the average daily energy, produced by the 4 PV that supply the SP. The average daily pumped water flow for the warm season was 12.8 m$^3$ h$^{-1}$, with values between 11.9 m$^3$ h$^{-1}$ in June and 13.3 m$^3$ h$^{-1}$ in August, respectively.

The volume of water pumped by SPAPV was directly dependent on the average duration of sunshine, determined as a daily monthly average, therefore the values were daily averages calculated for each month of the warm season. The average daily volume of water supplied during the vegetation period of the tomatoes, was 110.34 m$^3$ day$^{-1}$. The lowest values were in May, when only 93.47 m$^3$ day$^{-1}$ was collected, with the maximum

accumulated in August (131.27 m³ day⁻¹). For the arrangement of SPVPV with ADP, in the case of the WM variant, 3 watering times were required, in May and September, at an interval of approximately 10 days, and 9 watering times in June, with the interval between watering being shorter, of only 3 days (Table 9).

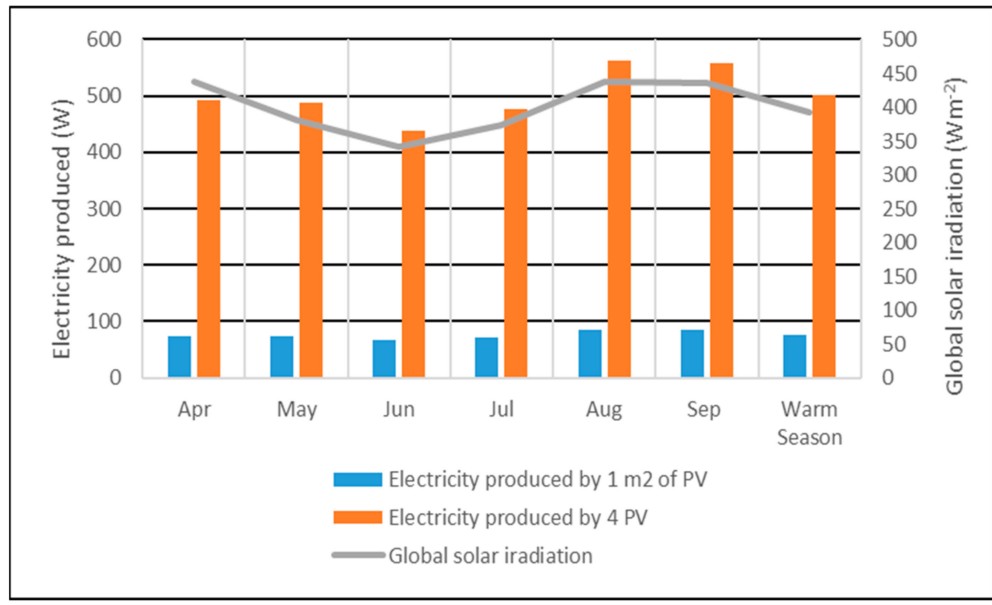

**Figure 8.** The photovoltaics (PVs) requirements for the operation of the solar pumps (SP).

**Table 9.** Number of solariums, that can be irrigated with ADP.

| Variant | Month | Pumped Flow | | n | m | Duration of Sunlight | Operating Time of Water Dispensers | Number of Irrigated Solariums |
|---|---|---|---|---|---|---|---|---|
| | | m³ h⁻¹ | L h⁻¹ | | L m⁻² | h | h | |
| WM | April | 12.7 | 12,700 | 3 | 18.2 | 7.36 | 4 | 1.84 |
| | May | 12.5 | 12,500 | 6 | 18.2 | 8.54 | 4 | 2.14 |
| | June | 11.9 | 11,900 | 9 | 18.2 | 8.17 | 4 | 2.04 |
| | July | 12.3 | 12,300 | 8 | 18.2 | 9.82 | 4 | 2.46 |
| | August | 13.3 | 13,300 | 6 | 18.2 | 9.87 | 4 | 2.47 |
| | September | 13.2 | 13,200 | 3 | 18.2 | 7.70 | 4 | 1.93 |
| MBF | April | 12.7 | 12,700 | 3 | 16.9 | 7.36 | 6 | 1.23 |
| | May | 12.5 | 12,500 | 7 | 16.9 | 8.54 | 6 | 1.42 |
| | June | 11.9 | 11,900 | 9 | 16.9 | 8.17 | 6 | 1.36 |
| | July | 12.3 | 12,300 | 10 | 16.9 | 9.82 | 6 | 1.64 |
| | August | 13.3 | 13,300 | 6 | 16.9 | 9.87 | 6 | 1.65 |
| | September | 13.2 | 13,200 | 3 | 16.9 | 7.70 | 6 | 1.28 |

Note: n—number of watering; m—watering norm; h—hours.

Given that in May the flow pumped by the SPAPV in one day was 93,772 L and the watering rate for a solarium is 4550 L, it covered the requirements for 2 solariums. Instead, due to the fact that the distribution at the plant is done with drippers that have to operate at 4 bar for 4 h, the second solarium could not be completely watered due to the fact that the operating time of the pump was 7.36 h. For the same reason, the number of whole solariums that could be fully irrigated for the WM variant in September was also one. In all the other months of the vegetation period, ADP could ensure for the WM variant the watering of two solariums. In the case of the MBF variant, the number of irrigated solariums was limited by the long duration of water distribution at the plant, which was done with the help of microtubes, with a required operating time of 6 h. In this case, the number of solariums that could be irrigated by this technique was only one for each month of the vegetation period.

For the arrangement with storage tanks (ATS), the water was distributed to the plant by gravity. This system allowed the extension of the collection and storage of pumped water, outside the period of water management by direct pumping. For the WM culture variant, the volume of water stored in one day was sufficient for the administration of two watering norms, i.e., two solariums, regardless of the month of the vegetation season (Table 10).

**Table 10.** Number of solariums that can be irrigated with a water storage tank (AST).

| Variant | Month | Pumped Flow | | n | m for a Solarium | Duration of Sunlight | Volume of Pumped Water | Number of Irrigated Solariums |
|---|---|---|---|---|---|---|---|---|
| | | $m^3\ h^{-1}$ | $L\ h^{-1}$ | | $m^3$ | h | $m^3\ Day^{-1}$ | |
| WM | April | 12.7 | 12,700 | 3 | 45.5 | 7.36 | 93.5 | 2.05 |
| | May | 12.5 | 12,500 | 6 | 45.5 | 8.54 | 106.8 | 2.34 |
| | June | 11.9 | 11,900 | 9 | 45.5 | 8.17 | 97.2 | 2.14 |
| | July | 12.3 | 12,300 | 8 | 45.5 | 9.82 | 120.8 | 2.65 |
| | August | 13.3 | 13,300 | 6 | 45.5 | 9.87 | 131.3 | 2.88 |
| | September | 13.2 | 13,200 | 3 | 45.5 | 7.70 | 101.6 | 2.23 |
| MBF | April | 12.7 | 12,700 | 3 | 42.25 | 7.36 | 93.5 | 2.21 |
| | May | 12.5 | 12,500 | 7 | 42.25 | 8.54 | 106.8 | 2.53 |
| | June | 11.9 | 11,900 | 9 | 42.25 | 8.17 | 97.2 | 2.30 |
| | July | 12.3 | 12,300 | 10 | 42.25 | 9.82 | 120.8 | 2.86 |
| | August | 13.3 | 13,300 | 6 | 42.25 | 9.87 | 131.3 | 3.11 |
| | September | 13.2 | 13,200 | 3 | 42.25 | 7.70 | 101.6 | 2.40 |

Note: n—number of watering; m—watering norm; h—hours.

For the MBF variant, due to the water losses being lower than in the case of WM, the water consumption of the tomatoes was lower and the watering norm was lower, resulting in a larger irrigated surface, but not exceeding two whole solariums. Therefore, the use of the arrangement with a water storage basin and gravitational distribution allowed for an increase in the efficiency of the water pumped by the SPAPV, provided that the water storage tank was installed at a height of at least 2.0 m and an ensured storage of an appropriate volume for a watering norm. This arrangement had the advantage that the watering norm could be administered outside the duration of sunlight, including at night with artificial light.

One way to increase the efficiency of water pumped by the SPAPV is through the combined use of the two possibilities for localized irrigation, ADP and ATS. On the day scheduled for watering, the water remaining from direct pumping should be stored in tanks for later administration. Considering that the administration of the watering norms can be advanced by one day or can be delayed by one day [94], in the case of ADP, it is possible to reach 2 watered solariums, and in the case of ATS, 4 served solariums.

Because the electricity produced by PV on days when no watering was scheduled was lost, we recommend adding to the arrangement scheme with accumulator batteries [49], which canlead to an increase in the number of irrigated solariums and increase possibilities for the use of the electricity produced for other utilities (lighting, heating, ventilation, etc.).

### 3.3. The Experimental Field from Husasău de Tinca

Tomato yields grown organically in drip-irrigated solariums from Husasău de Tinca to cover the optimal water consumption (ETRo) had average values (2015–2017) of 86.61 T $ha^{-1}$ for the WM variant and 129.83 T $ha^{-1}$ for the MBF variant. The average yield increase was 29.83% for the MBF variant compared to the WM variant (Table 11).

**Table 11.** Organic tomato yields obtained in the solariums from Husasău de Tinca (2015–2017).

| Variants | Years | | | | | | | | | | | | Average (2015–2017) | | | |
|---|---|---|---|---|---|---|---|---|---|---|---|---|---|---|---|---|
| | 2015 | | | | 2016 | | | | 2017 | | | | | | | |
| | T ha$^{-1}$ | % | ± | Sign | T ha$^{-1}$ | % | ± | Sign | T ha$^{-1}$ | % | ± | Sign | T ha$^{-1}$ | % | ± | Sign |
| WM | 84.63 | 100.00 | - | - | 87.97 | 100.00 | - | - | 87.23 | 100.00 | - | - | 86.61 | 100.00 | - | - |
| MBF | 113.60 | 134.24 | 28.97 | *** | 112.03 | 127.35 | 24.06 | *** | 111.72 | 128.08 | 24.49 | *** | 112.45 | 129.83 | 25.84 | *** |
| | | | | | | Fischer test (F) | | | | | | | | | | |
| F 5% | 5.79 | $F_C$ [1] = | 2503 | | 5.79 | $F_C$ [1] = | 1472 | | 5.79 | $F_C$ [1] = | 713.8 | | 3.55 | $F_C$ [1] = | 175.9 | |
| F 1% | 10.92 | | | | 10.92 | | | | 10.92 | | | | 6.01 | | | |
| | | | | | | Student test (t) | | | | | | | | | | |
| LSD [2] 5% | | | 0.685 | | | | 0.832 | | | | 1.882 | | | | 0.971 | |
| LSD [2] 1% | | | 1.037 | | | | 1.260 | | | | 2.850 | | | | 1.323 | |
| LSD [2] 0.1% | | | 1.667 | | | | 2.024 | | | | 4.579 | | | | 1.813 | |

[1] The calculated Fischer value; [2] Limit of the significant difference; *** Production differences are very statistically significant ($p > 0.01$).

The yields in the years of cultivation varied between 84.63 T Hha$^{-1}$ (2015) and 87.97 T ha$^{-1}$ (2016) in the WM variant and between 111.72 T ha$^{-1}$ (2017) and 113.6 T ha$^{-1}$ (2015) in the MBF variant. The relative increases in yield were between 27.35% and 34.24%. The MBF variant brought annual yield increases of 24.06–28.97 T ha$^{-1}$, very significant from a statistical point of view, at the level of accuracy $p > 0.01\%$. Additionally, the average yield increase of the MBF variant for the analyzed period of 25.84 to ha$^{-1}$ was very statistically significant at the same level of accuracy ($p > 0.01\%$).

### 3.4. Effectiveness of Measures to Reduce the Effects of Climate Change and Energy Consumption

The economic efficiency was assessed through the specific investment required for the arrangement of localized irrigation with ADP and ATS using tomatoes yields made in 2016 by the two crop systems studied (WM and MBF). Specific investment (SI), calculated in RON and Euro at the parity of 2020 (1 Euro = 4.84 Euro) was 1,039,610 Lei ha$^{-1}$ (214,795 Euro ha$^{-1}$) in the case of ADP, and respectively 982,470 Lei ha$^{-1}$ (202,990 Euro ha$^{-1}$) in the case of ATS (Figure 9).

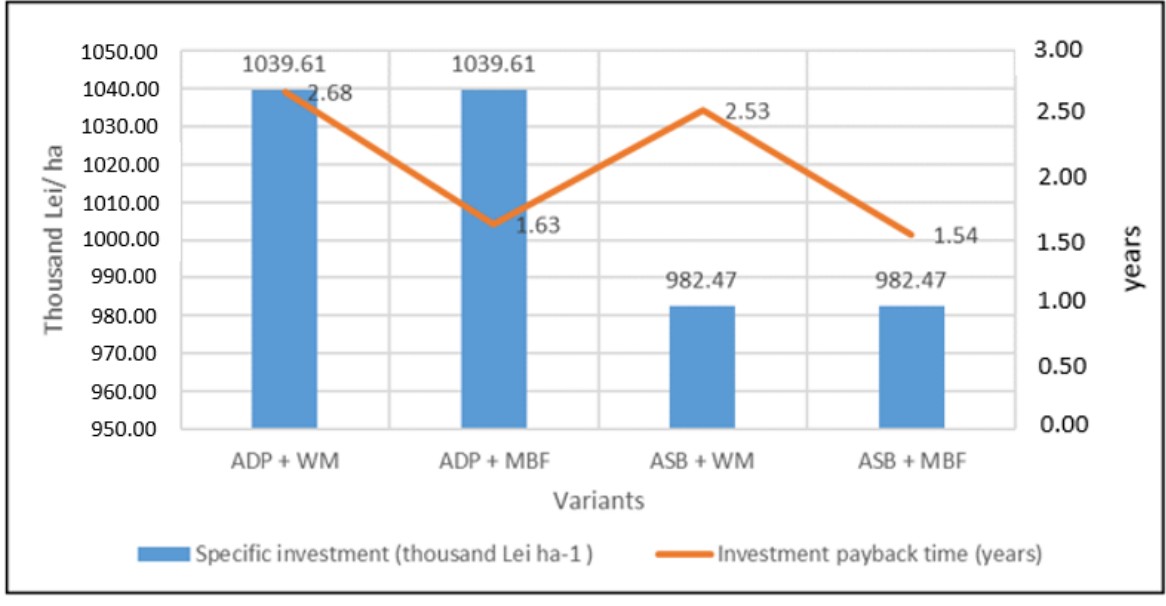

**Figure 9.** Indicators of economic efficiency.

IPT was longer for WM, at 2.68 years for ADP and 2.53 years for ATS, respectively. IPTs were lower in the MBF variant, 1.54 years at ATS due to higher yields and lower SI than at ADP and 1.63 years for ADP.

The indices of water use and irrigation water us, in Husasău de Tinca were assessed for the 2016 year. WUC values showed that the water consumed by irrigated tomatoes was higher in the WM version than in the MBF version, but it was used more efficiently in the MBF version, requiring only 0.059 m$^3$ of water to produce 1 kg of tomatoes, while in the WM version, 1 kg of tomatoes required 0.059 m$^3$ of water consumed (Table 12).

**Table 12.** The indices of water use and irrigation water use in Husasău de Tinca.

| Variant | ETRo | Σm | Y | WUC (ETRo/Y) | WUE (Y/ETRo) | IWUC (Σm/Y) | IWUE (Y/Σm) |
|---|---|---|---|---|---|---|---|
| | M$^3$ ha$^{-1}$ | M$^3$ ha$^{-1}$ | Kg ha$^{-1}$ | M$^3$ kg$^{-1}$ | Kg m$^{-3}$ | M$^3$ kg$^{-1}$ | Kg m$^{-3}$ |
| WM | 6945.7 | 6370 | 87,970 | 0.079 | 12.67 | 0.072 | 13.81 |
| MBF | 6594.0 | 6422 | 112,030 | 0.059 | 16.99 | 0.057 | 17.44 |
| Differences | | | | +0.02 | −4.32 | +0.015 | −3.63 |

The IWUC values were 0.072 m$^3$ kg$^{-1}$ in the WM variant, showing that in order to produce 1 kg of tomatoes, 0.073 m$^3$ of water administered by irrigation was needed, while in the MBF variant, 0.015 m$^3$ less was needed.

WUE indicated that in the WM version, 1 m$^3$ of water consumed by irrigation in optimal conditions, corresponded to a tomato yield of 12.67 kg, while in the MBF version, it was 4.32 kg higher. IWUE, the ratio between yield and water administered by irrigation, showed that for 1 m$^3$ of water, 13.81 kg of tomatoes were obtained, for the WM variant. In the case of the MBF variant, the efficiency of water use was higher, obtaining 17.44 kg of tomatoes for each m$^3$ of water administered by irrigation. The efficiency of water use was higher for the MBF variant, both for the optimal water consumption (+4.32 m$^3$ kg$^{-1}$) and for the irrigation norm (+3.63 m$^3$ kg$^{-1}$) administered, indicating the superiority of this measure of water conservation in the soil.

## 4. Discussion

### 4.1. Irrigation Regime of Tomatoes Grown in Solariums

Comparing the ETRo values of tomatoes obtained starting from the assurance of the air temperature of 80%, with those obtained in the period 1999–2001, using the Piche evaporimeter [70], they are lower than 7153.1 m$^3$ ha$^{-1}$ for WM and 6958.3 m$^3$ ha$^{-1}$ respectively, for MBF.

If we take into account, the ETRo of tomatoes grown in solariums estimated in 2018, starting from the insurance of not exceeding the precipitation of 80% [74], they are higher than those determined from the insurance air temperature, with: 597.5 m$^3$ ha$^{-1}$ for the WM variant and 668.4 m$^3$ ha$^{-1}$, respectively, for the MBF variant.

Therefore, estimating the water consumption of tomatoes grown in the solarium with the help of temperatures leads to lower values of ETRo and allows the forecast of watering, starting from the measurement of air temperature inside the solarium. We believe that this design method leads to water saving, even in the conditions of the current trend of increasing temperatures during the vegetation period of crops.

Tomatoes grown in the field in Szarvas (Hungary) in pedoclimatic conditions were very close to ours in the period 2017–2019, and consumed an average of 3949.7 m$^3$ ha$^{-1}$, of which 2170.0 m$^3$ ha$^{-1}$ were covered by precipitation and soil water supply [95]. For the conditions in the Crișurilor Plain from Husasău de Tinca, the average consumption of tomatoes grown in solariums (1999–2001) was 7090 m$^3$ha$^1$, of which 5180 m$^3$ ha$^{-1}$ were covered by irrigation, and the remaining 269 m$^3$ ha$^{-1}$ by soil water reserve [74].

Under these conditions, the average irrigation rate administered was 1779.7 m$^3$ ha$^{-1}$, much different from the one administered by us in the solarium, of 6370–6422 m$^3$ ha$^{-1}$.

The average ETRo (2000–2002) of sun-grown tomatoes in Cluj-Napoca, an area characterized by lower temperatures than those in our experimental field, was between 4232.8

and 4821.9 m$^3$ ha$^{-1}$ [96]. These values suggest that as the temperature of the air rises, the water consumption of tomatoes grown in the solarium increases.

The estimated water consumption of tomatoes grown in the greenhouse by measuring the flow of sap under stress was between 2234.1 and 2921.8 m$^3$ ha$^{-1}$ [75]. These very low values compared to those established by us are due to the fact that the applied irrigation regime did not cover the optimal consumption (ETRo) in order to stress the crop to measure the sap flow.

Li, 2022, in their research on the optimization of the irrigation strategy with a regulated deficit for greenhouse tomatoes, use as a control to compare the effect of irrigation norms that induce water stress, the value of 328.6 mm, which represents 3286 m$^3$ ha$^{-1}$ [94]. This value of ETRo is closer to that established by us in the experimental field for tomatoes grown in solarium.

However, we believe that this design method, based on the air temperature measured inside the solarium, saves water in relation to the optimal consumption (ETRo) of tomatoes, even in the current trend of rising temperatures.

### 4.2. The Main Characteristics of SPAPV from Tomatoes Grown in Solariums

The reduction of the volumes of water for feeding the tomatoes grown in solariums, in the conditions of supplying the pumps of the drip irrigation system, with electricity from the network, leads to energy savings and implicitly to the reduction of production costs.

If we compare the values obtained by estimating global irradiation, using the Photovoltaic Geographical Information System [83] with those measured in Timisoara, we notice that the values obtained for Oradea are very close, these being in Oradea 342.4–438.2 W m$^{-2}$ and in Timișoara of 351.7–458.2 W m$^{-2}$.

Cervera-Gascó et al. 2022, show that in the conditions of global climate change, characterized by frequent variations, the SPAPVs used for irrigating crops, in the established model for estimating the energy generated, have very small errors. This is a great advantage for watering forecasting and scheduling possibilities [97].

The use of SPAPVs for irrigation has the advantage that it works with green, renewable electricity, with special effects in the process of reducing greenhouse gas emissions from agriculture [98].

Given the current energy crisis and the photovoltaic potential of the Crișurilor Plain, by using the water pumping of the SPAPVs, the energy independence of the irrigation system is obtained and the energy costs are completely eliminated.

### 4.3. The Yields in Experimental Field from Husasău de Tinca

Average yields achieved in the solariums in the experimental field from Husasău de Tinca in the period 2015–2017, when the average temperature of the vegetation period was 18.92 °C, are different from those for the period 1999–2001 when the temperature was 18.21 °C. The yield was lower in the WM variant (86.61 T ha$^{-1}$ vs. 87.9 T ha$^{-1}$) and higher by 0.85 T ha$^{-1}$ in the MBF variant, suggesting the superior efficiency of mulching of the tomatoes grown.

It is noteworthy that these yields were obtained in the conditions of a culture of organic tomatoes, which implies a total lack of treatments with plant protection products and chemical fertilizers.

Research on field-grown tomatoes with different drip irrigation standards has shown that under appropriate agrotechnical conditions, production increases in proportion to the irrigation rate administered, reaching up to 90–110 T ha$^{-1}$ [95].

The average yield of tomatoes for a period of three years (2000–2002) grown in solariums, drip-irrigated with irrigation norms to cover the optimal consumption, in the area of the Someșan Plateau (where the air temperature is lower than in Crișurilor Plain), was 78.7 T ha$^{-1}$, lower than those achieved in Husasău de Tinca [96].

Research conducted in Brazil on various materials used for mulching tomatoes has shown an increase in soil temperature in the case of dark plastic films. One might think

that these temperature increases lead to an increase in the water consumption of tomatoes and thus to a reduction in production. However, mulching with colored plastic gaps led to an increase in production by 33–34 compared to the control variant, without mulch and without weed control [99]. This research confirms our conclusions that mulching reduces the amount of water lost through evaporation and not the specific consumption of the crop perspiration.

*4.4. Effectiveness of Measures to Reduce the Effects of Climate Change and Energy Consumption*

The costs for arranging one hectare of ADP are higher than for ATS by 57,140 Lei, because in the second case, it is possible to give up various safety devices (flow meter and manometer) and the tank for fertilizer distribution (the tank for water storage can be used).

The ITPs for the analyzed SPAPV variants exceed 2 years, both for the ADP variant and for the ATS variant, in the case of cultivating tomatoes without mulching (WM) due to the lower ANP value than in the MBF variant. The application of the method of conserving the water reserve from the soil by mulching, determines higher tomato yields, higher ANPs than in the WM variant, and implicitly ITPs of SPASPV less than 2 years.

IWUC values are lower than WUC values, because part of the optimal water consumption (ETRo) is done by the irrigation norm ($\Sigma$m) supplemented by the soil water reserve. In both variants, the values of the coefficients are lower for MBF, showing that the water consumed, regardless of whether it is optimal consumption (ETRo) or irrigation water, was preserved by the soil more efficiently than in the case of WM.

For the cultivation conditions of tomatoes irrigated in solariums in Cluj Napoca (2000–2002), an average value of IWUC of 0.058 m$^3$ kg$^{-1}$ was reported [74], which is very close to the that obtained by us for the MBF variant (0.057 m$^3$ kg$^1$). This lower value shows more efficient use of irrigation water.

Research on the influence of different methods of soil aeration and plastic film mulching of field-grown tomatoes on IWUE reports values between 14.8 and 17.7 kg m$^{-3}$ [100]. The values are close but still higher than those obtained by us in the solarium, and are due to the application of soil aeration methods that lead to increased water storage capacity in the soil.

The IWUE values obtained with the help of SPAPV in 2016 are higher than those determined in 2018 of 10.2 kg m$^{-3}$ for WM and 12.3 kg m$^{-3}$ for MBF, respectively.

These show higher tomato yields for the same amount of water administered by irrigation. The large differences between the IWUE values can also be explained by the higher annual precipitations and average temperatures in 2018 (702.2 mm, 12.45 °C) than in 2016, (693.2 mm, 10.93 °C) which determined the coverage of a higher percentage of water consumption of tomatoes from the soil water reserve.

**5. Conclusions**

The average annual temperature with the assurance of exceeding 20% at the Oradea Meteorological Station of 12.2 °C, was registered in the agricultural year 2015–2016. For the vegetation period of tomatoes grown in solariums, the average air temperature was 18.8 °C, while inside solariums it was 22.8 °C. PET has monthly values between 75.31 mm in April and 171.46 mm in July.

The amount of water needed to cover the water consumption of tomatoesin the case of the WM variant of 6370 m$^3$ ha$^{-1}$, requires 35 waterings, and for MBF 6422 m$^3$ ha$^{-1}$, it is administered in 38 waterings.

Horizontal global solar irradiation, accumulated during the vegetation period of tomatoes grown in solariums (April–September) is 1005.3 kWh m$^{-2}$, while in the case of PVs inclined at 45°, it is estimated at 1038.9 kWh m$^{-2}$. The average hourly value is 391.8 W m$^{-2}$, with monthly values between 342.4 W m$^{-2}$ in June and 438.2 W m$^{-2}$ in August.

The SPAPV used is composed of the SP produced by Lorentz, PS600 BADU Top12, which is delivered together with the PS 600 controller and 4 PVs AE320M6-60, produced in Germany, with supplied power of 320 W at a yield of 19.3%.

The average daily flow pumped by the SPAPV for the vegetation season is 12.8 m$^3$ h$^{-1}$, with values between 11.9 m$^3$ h$^{-1}$ in June and 13.3 m$^3$ h$^{-1}$ in August, respectively. The average daily water volume is 110.34 m$^3$ day$^{-1}$, with the lowest values in May when only 93.5 m$^3$ day$^{-1}$ are collected, with the maximum accumulated in August (131.3 m$^3$ day$^{-1}$).

In the case of ADP, the WM variant can water 2 entire solariums, except in September when only one is watered. ADP ensures for the MBF variant the watering of 2 whole solariums for the entire vegetation period. ATS increases the efficiency of irrigation water use and it is possible to water 2 whole solariums for both variants analyzed. By combining the two possibilities of localized irrigation, ADP and ATS, by storing the pumped water in tanks and by administering the watering norm in advance or late by one day, it is possible to reach 4 solariums served.

SI is 1039.6 th Lei ha$^{-1}$ (214.8 th Euro ha$^{-1}$) in the case of ADP and 982.5 th Lei ha$^{-1}$ (203.0 th Euro ha$^{-1}$) in the case of ATS. The IPT for the WM variant is between 2.53 years for ATS and 2.68 years for ADP, respectively. The IPTs are, in the case of the MBF variant, 1.54 years for ATS and 1.63 years for MBF and ADP, respectively.

IWUC values show that irrigation water is used more efficiently in the case of the MBF variant, the volume of irrigation water required to produce a ton of tomatoes is 57.3 m$^3$ and 72.4 m$^3$ in the WM variant. The IWUE values show that for 1 m$^3$ of water administered by irrigation, 13.8 kg of tomatoes are obtained for the WM variant and 17.4 kg of tomatoes for each m$^3$ of water administered, for the MBF variant.

For the pedoclimatic conditions in the Crișurilor Plain, the measures taken to reduce the effects of global climate change and to save electricity are economically efficient in terms of the efficiency of irrigation water use.

In order to save water and electricity, in the case of tomatoes grown in solariums, the arrangement for localized irrigation with a water storage tank (ATS) and the culture variant with mulching using black foil (MBF) is recommended.

**Author Contributions:** Conceptualization, M.C. (Mihai Cărbunar) and N.C.S.; methodology, O.M.; software, M.C. (Mircea Curilă); validation, I.B., A.S. and T.V.; formal analysis, A.V.; investigation, A.P. and M.L.C.; data curation, C.O.; writing—original draft preparation, N.C.S. and M.C. (Mihai Cărbunar); writing—review and editing, M.L.C.; supervision, N.C.S. All authors have read and agreed to the published version of the manuscript.

**Funding:** This research received no external funding.

**Institutional Review Board Statement:** Not applicable.

**Informed Consent Statement:** Not applicable.

**Data Availability Statement:** Not applicable.

**Acknowledgments:** The authors thank Sorin Curilă (University of Oradea, Faculty of Electrical Engineering and Information Technology) for technical support in the design of the solar pump and solar panels.

**Conflicts of Interest:** The authors declare no conflict of interest.

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
