# Peer review of "Effectiveness of Measures to Reduce the Influence of Global Climate Change on Tomato Cultivation in Solariums—Case Study: Crișurilor Plain, Bihor, Romania"

_agriculture, doi:10.3390/agriculture12050634_

Round 1

Reviewer 1 Report

Dear authors,

The manuscript evaluated "Effectiveness of measures to reduce the influence of global climate change on tomato cultivation in solariums. Case study:  CriÈ™urilor Plain, Bihor, Romania”. The topic is interesting, in particular taking into account climate change and food security. A lot of work is done, and the methodology used is adequate to the objective of the study. However, some revisions are still required as shown below.

Abstract

The novelty of this study and a solid conclusion regarding the obtained results should be given.

Introduction

The study hypothesis should be improved. The significance of the study and a solid hypothesis of the present study at the end of the introduction should be provided to give the reader more information regarding the purpose of the study and the mechanistic used to achieve this objective.

The introduction should also include the recent literature discussing the research hypothesis. Please include the related recent literature.

The references should be up to date in the whole text.

Methods

The methods should be written in more detail to be reproducible.

Discussion

The discussion should be interpreted with the results as well as discussed in relation to the present literature.

The references section should include studies about the recent literature presented on this topic as well.

Kind regards!

Author Response

Cover Letter – Reviewer I

            Thank you, from the bottom of my heart, for your efforts in reviewing our proposed paper for publication in Agriculture. Your comments were pertinent and very helpful.

            Changes made as a result of your comments are presented together with those of the second reviewer.

            At your suggestion, we've made the following improvements:

  • in the abstract was mentioned the main novelty brought by our work;
  • the conclusions were reformulated on a more solid basis, given the mention of the hypotheses of the study;
  • working hypotheses were mentioned at the end of the Introduction;
  • recent literature on hypotheses has been added to references;
  • although the literature originally cited was very new (54.4% from 2020-2022) it has been improved, the percentage of works in the last 3 years being 55%;
  • regarding the methodology, it is developed in detail, being presented all the calculation relations; In addition, the mechanism used to achieve the objectives is clearly shown in Figure 1;
  • discussions were developed by comparing our results with those in the recent literature;
  • English expression and grammar have been corrected.

I am sending you both the work with the changes marked in "Track Modification" and the final version resulting from the correction.

I hope that the last form of our study will allow its publication in the prestigious journal Agriculture.

Thanks again for your support.

With best wishes,

Prof. Nicu Cornel Sabău

University of Oradea, Romania

Faculty of Environmental Protection

Department of Environmental Engineering

Cover Letter – Reviewer I

            Thank you, from the bottom of my heart, for your efforts in reviewing our proposed paper for publication in Agriculture. Your comments were pertinent and very helpful.

            Changes made as a result of your comments are presented together with those of the second reviewer.

            At your suggestion, we've made the following improvements:

  • in the abstract was mentioned the main novelty brought by our work;
  • the conclusions were reformulated on a more solid basis, given the mention of the hypotheses of the study;
  • working hypotheses were mentioned at the end of the Introduction;
  • recent literature on hypotheses has been added to references;
  • although the literature originally cited was very new (54.4% from 2020-2022) it has been improved, the percentage of works in the last 3 years being 55%;
  • regarding the methodology, it is developed in detail, being presented all the calculation relations; In addition, the mechanism used to achieve the objectives is clearly shown in Figure 1;
  • discussions were developed by comparing our results with those in the recent literature;
  • English expression and grammar have been corrected.

I am sending you both the work with the changes marked in "Track Modification" and the final version resulting from the correction.

I hope that the last form of our study will allow its publication in the prestigious journal Agriculture.

Thanks again for your support.

With best wishes,

Prof. Nicu Cornel Sabău

University of Oradea, Romania

Faculty of Environmental Protection

Department of Environmental Engineering

Reviewer 2 Report

the manuscript entitled " Effectiveness of measures to reduce the influence of global climate change on tomato cultivation in solariums. Case study: CriÈ™urilor Plain, Bihor, Romania." needs to be improved by shortening introduction (trying not to omit important parts) and improving the discussion section. some sentences are too long or difficult to understand, read the manuscript once more and improve them.  

Considering the importance of the subject (climate changes) and need for cultivation in different conditions (like solariums) and energy and water managments, the manuscript could be published after being revised (major revision).

The manuscript needs English revising since many sentences are too long or sometime complicated therefore difficult to understand like lines 56-59 (difficult to get the main subject), line 59-61, 73-76, 109-112, 126, 149-152,155-157, 343,344,346,372-373, 436-437, 454, 495-496, 501 , 515-517,577-583,611 (similar problem),  422-424, 493-494-515-522,580 (font), 430-432, 439, 445-486, 501-503, 507-513,  whole results, (grammar - time usage), 515,562,570 (grammar) and some others.... In addition, I found the discussion part insufficient due to lack of enough comparing to other studies and giving enough reasons for some parts like 628,654-661.....

If the authors meant to have only a recommendation for cultivation in solariums with the lower water and energy costs, the discussion could be considered enough. however, using other studies could give more clear results, as well. some discussion parts have similar problem considering verb tense, complicated sentences..... the authors gave one page for discussion part that could be enhanced by using other studies or more explanations to have it a real discussion face not  a final results appearance. the authors used 90 references for the paper that seems too high for a research paper particularly concerning poor discussion. the attached is the MS highlighted some problems.

Author Response

Cover Letter – Reviewer II

            Thank you, from the bottom of my heart, for your efforts in reviewing our proposed paper for publication in Agriculture. Your comments were pertinent and very helpful.

            Changes made as a result of your comments are presented together with those of the second reviewer.

            At your suggestion, we've made the following improvements:

  • although the first reviewer suggested adding at the end of the introduction the working hypotheses I reduced this chapter by removing two large paragraphs;
  • for a better understanding of English, it has been corrected according to all your suggestions;
  • discussions were developed by comparing our results with those in the recent literature;
  • the conclusions have been improved to make them clearer by presenting working hypotheses
  • although the number of references was high (90), of which 54.4% in the last 3 years, it could not be reduced;
  • due to the requirement of the first reviewer to introduce the working hypotheses and due to the development of the Discussions chapter, the number of references has increased to 100 of which 55% are recent, from 2020-2022.

I am sending you both the work with the changes marked in "Track Modification" and the final version resulting from the correction.

I hope that the last form of our study will allow its publication in the prestigious journal Agriculture.

Thanks again for your support.

With best wishes,

Prof. Nicu Cornel Sabău

University of Oradea, Romania

Faculty of Environmental Protection

Department of Environmental Engineering

Round 2

Reviewer 1 Report

Dear authors,

You have addressed all the comments.

Kind regards,

Reviewer 2 Report

the comments were approximately applied on the MS. however some parts related to English part needs to be checked once again like lines 67-71 including spelling "Which"

please read the MS once more for possible mistakes and correct all.